# Probiotic-Based Sanitation in the Built Environment—An Alternative to Chemical Disinfectants

**Ashley M. Ramos and Aubrey L. Frantz ***

Department of Natural Sciences, University of North Texas at Dallas, Dallas, TX 75241, USA;
ashleyramos2@my.untdallas.edu
* Correspondence: aubrey.frantz@untdallas.edu

**Abstract:** The use of conventional chemical disinfectants is a common practice in built environments and has drastically increased in response to the COVID-19 pandemic. While effective for instantaneous disinfection, the application of chemical disinfectants to indoor surfaces is associated with recontamination and is prone to select for antimicrobial-resistant pathogens. In contrast, probiotic-based sanitation (PBS) relies on the premise that probiotic bacteria, namely apathogenic *Bacillus* spp., when combined with eco-friendly detergents and applied to indoor surfaces can outcompete and exclude pathogens. Recent in situ studies assessing PBS in healthcare settings have demonstrated overwhelmingly positive results, including significant reductions in pathogen burden, antimicrobial-resistant genes and nosocomial infections, yet these studies are limited in duration and scope. Here, we review results of *Bacillus*-based PBS in practice, identify knowledge gaps and discuss the considerations for the widespread use of PBS in built environments. In a time when indoor cleaning and disinfection has come to the forefront, PBS may offer an attractive, effective and sustainable alternative to conventional chemical disinfectants.

**Keywords:** disinfectant; probiotic-based sanitation; antimicrobial resistance; COVID-19; *Bacillus*; microbiome; built environment

## 1. Introduction

The use of chemical disinfectants is a common practice in our built environments (BE), including homes, workplaces, schools, industries, public transportation, healthcare clinics and hospitals. While hundreds of different antimicrobial chemicals are used as the active ingredients in disinfectants, the most common antimicrobial chemical groups include quaternary ammonium compounds (QAC), chlorine and chlorine compounds, alcohols and phenols [1,2]. In 2020, the Environmental Protection Agency (EPA) released its initial List N: Disinfectants for Use Against SARS-CoV-2, which now contains over 500 chemical disinfectant products that meet the EPA's criteria for use against SARS-CoV-2 [3]. The production and consumer use of these chemical disinfectants rose drastically during the COVID-19 pandemic and is predicted to substantially increase over the next decade [4].

While effective and consistent sanitation practices are essential to protect individual and public health, conventional chemical-based disinfection has several notable limitations that are directly and indirectly associated with adverse human health effects. Chemical disinfectants are time dependent and are often only effective on cleaned surfaces for minutes to hours after proper application; therefore, these products are generally not effective against recontamination [5,6]. Persistent recontamination is a leading cause of hospital acquired infections (HAIs), and contributes significantly to the spread of infection within childcare centers and households [7–9]. Additionally, recent studies have identified chemical disinfectants as a leading cause of bacterial resistance to disinfectants and cross-resistance to antibiotics [10–12]. The effective dose of any biocide depends on the formulation, the type of surfaces treated, temperature and contact time [13,14]. Prolonged exposure to sublethal concentrations of chemical disinfectants can select for tolerant organisms, and therefore

simultaneously increase the abundance of antimicrobial-resistant genes and antibiotic-resistant organisms [15]. In fact, the environments with the highest levels of disinfectant use also harbor the highest rates of multi-drug-resistant microbes [16,17]. Nearly 10% of bacterial samples isolated from local indoor fitness centers frequently disinfected with QAC-containing products were found to be resistant to in-use disinfectant concentrations and the majority of QAC-resistant strains were also resistant to ampicillin, erythromycin, penicillin, ciprofloxacin and chloramphenicol [18]. Survival of QAC-tolerant MRSA on automated teller machines was suggested to be facilitated by low disinfectant concentrations [19], and numerous outbreaks have been documented and attributed to disinfectant-resistant pathogens that have contaminated antiseptic products [20–23]. Antimicrobial resistance has been declared as one of the top ten global public health threats that require urgent multisectoral action [24].

The prolific and unprecedented use of chemical disinfectants during the COVID-19 pandemic has also led to a rise in acute exposure incidents [25–28]. Consistent with public health recommendations for increased cleaning, 2020 survey results indicate that more than 70% of households increased the frequency of disinfection during the COVID-19 pandemic, and 80% of the households reported routinely using chemical disinfectant products [29]. Unfortunately, more than one third of households use chemical disinfectants unsafely [30,31]. Comparing data from the US National Poison Data System from January through March of 2019 and 2020, National Poison Control Centers received a 20.4% increase in calls related to disinfectant exposures in 2020 [25]. Similar data have been reported internationally [27,28]. Moreover, the frequent detection of QACs in blood, tissue and breastmilk confirms that human exposure to chemical disinfectants is significant and widespread [29,32,33]. Notably, QAC concentrations in blood samples taken in 2020 were correlated with biomarkers of inflammation, mitochondrial dysfunction and sterol imbalance [34]. Numerous case studies have suggested that acute exposure to chemical disinfectants can cause irritant and allergic contact dermatitis and contribute to the development of occupational asthma [35–38]. A multi-year study from 2009 to 2015 identified regular chemical disinfectant use by female nurses as a major risk factor for developing COPD [39]. Likewise, chlorine compounds from chlorine-based disinfectants have been known to react with organic materials in water to form toxic and corrosive disinfection by-products that are harmful to humans, animals, plants, ecosystems and built environments [40]. Several reviews have attempted to evaluate current consumer exposure to chemical disinfectants and associated health risks and have concluded that indoor chemical disinfectant use poses distinct human and environmental health threats [40–43]

Given the evidence, there is a pressing need for effective, safe and eco-friendly sanitation approaches as an alternative to chemical disinfectants. Along with the growing understanding of the importance of microbiome health, emerging evidence supports a bidirectional hygiene ("bygiene") approach to sanitation that aims to target and reduce pathogen burden yet maintains the natural microbial biodiversity in the built environment—thus, limiting the risk of infection and promoting the survival of, and exposure to, beneficial microbes [44–46]. In contrast, the sole aim of disinfection is to eliminate pathogenic microorganisms on objects and surfaces [2]. In the process, chemical disinfectants offer a trade-off—eradicate the potential pathogen, but also destroy the resident microbes. A recent assessment of disinfectant versus plain soap sanitation approaches found that the use of chemical disinfectants strongly promoted the survival of pathogenic bacteria on cleaned surfaces [47]. Specifically, on surfaces cleaned with chemical disinfectants, pathogens outcompeted the resident microbes, but on surfaces cleaned with plain soap, resident microbes outcompeted the pathogens [47]. These results suggest that both maintaining microbial diversity and fostering an environment where apathogenic species can outcompete pathogens are critical components of effective sanitation systems.

This competitive exclusion principle is the basis for biocide-free, probiotic-based sanitation (PBS). The benefits of probiotics on human health have been increasingly touted for years [48–50]. Probiotics, defined as "living microorganisms that confer a benefit to the

host when administered in adequate amounts" [48], have been widely accepted as safe and effective therapies to prevent and treat a wide range of human health problems including inflammatory bowel disease (IBD), antibiotic-resistant skin infections, gastrointestinal and urogenital bacterial infections, gingivitis and allergic disorders [51–57]. The mechanisms by which probiotics provide these human health benefits are still actively being investigated. The prevailing theory is that probiotic species, in adequate numbers, are able to outcompete pathogens for nutrients and space via a variety of microbial mechanisms (production of antimicrobial compounds, community interactions, quorum sensing, host immune system modulation, etc.); although the particular mechanisms of action and host effects are largely species- or even strain-specific [58].

Similar to individuals, built environments (BE) have microbiomes that are composed of microorganisms inhabiting surfaces and inanimate objects, and circulating in ventilated air [59,60]. BE microbiomes are generally composed of commensal microbes of human origin that colonize and persist in indoor environments. The composition of BE microbiomes is highly dynamic and is influenced by human and environmental factors including climate, temperature, sanitation practices, level of human occupancy and various other human activities [61]. Recent studies indicate BE microbiomes actively impact human and public health, acting as a potential reservoir for pathogens [62,63]. A global, 3-year, 60-city study that cataloged the microbiomes of mass-transit systems identified the presence of antimicrobial resistance (AMR) genes in the majority of sampled locations, although unevenly distributed [64]. Routine disinfectant use has been suggested to promote the survival and proliferation of antimicrobial-resistant microbes in BEs [63]. Accordingly, the idea that PBS could be used to effectively and sustainably modulate and protect BE microbiomes without the prolific use of chemical disinfectants has become an attractive, yet underacknowledged, sanitation strategy in recent years. This general PBS hypothesis was first proposed in 2009, with the aim of combating nosocomial infections by applying probiotics to frequently contaminated patient equipment [65]. Since then, eco-friendly detergents with high concentrations of apathogenic, food-grade bacterial spores of the *Bacillus* genus (namely *B. subtilis*, *B. pumilus* and *B. megaterium* species) have been the predominant PBS system investigated. *Bacillus* spores are effectively able to survive a wide range of temperatures and pH environments [66,67]. Once diluted and applied, *Bacillus* spores are able to germinate and quickly colonize dry inanimate surfaces, ultimately dominating the composition of the resident microbiome and competitively excluding pathogens from surviving and colonizing these high-touch surfaces [68]. Here we review recent in situ studies assessing the effectiveness of *Bacillus*-based PBS in practice, an approach which is predominantly limited to European hospital settings. We identify knowledge gaps in our understanding of the potential short and long-term effects of PBS and discuss the considerations for the widespread use of PBS in built environments.

## 2. Materials and Methods

Pubmed, Google Scholar, Embase and EBSCOhost databases were used to identify peer-reviewed, primary research articles investigating the application of *Bacillus*-based probiotic sanitation systems in built environments. The key terms searched in these databases were as follows: "*Bacillus*" AND "probiotic-based", AND "sanitation OR disinfection". The searches produced 233 results. All search results were reviewed and the studies comparing the in situ effectiveness of *Bacillus*-based PBS to conventional chemical disinfectants were selected for further evaluation. Non-peer-reviewed articles were excluded. Only in situ studies were included in the Results—original articles that produced entirely in vitro results, or were limited to laboratory settings, were excluded. The same inclusion and exclusion criteria were used for all databases. After critical review, 15 results were identified as original research articles that assessed *Bacillus*-based, probiotic-based sanitation systems in built environments.

## 3. Results

Early in vitro studies carried out in 2013 and 2014 investigating the effectiveness of *Bacillus*-based PBS on hospital-like hard surfaces demonstrated a striking 80–99.9% average reduction in pathogen burden within one day of application [69,70]. Importantly, this reduction in pathogen burden remained stable for over five days, in contrast to the limited biocidal efficacy of chemical disinfectants, which waned significantly within one hour of treatment [70]. Yet, in the past decade, there have been fewer than twenty published peer-reviewed studies investigating the efficacy of PBS in practice (Table 1). Many of these in situ studies were conducted in European hospitals and healthcare settings and used the patented *Bacillus*-based probiotic cleaning and hygiene system developed by Copma (Ferrara, Italy) [8,69,71–77]. All studies identified in Table 1 compared the effectiveness of PBS to conventional chemical disinfectants.

**Table 1.** Outcomes of Probiotic-Based Sanitation (PBS) in Practice.

| Reference | Built Environment | Location | Duration | Outcome (PBS vs. Chemical Disinfectants) |
|---|---|---|---|---|
| Vandini et al. (2014) [69] | One in-patient and one out-patient general medicine ward (samples collected from the corridor floor, room floor, toilet and sink) | Ferrara, Italy | 4 months | 80% reduction in pathogen burden (*S. aureus*, coliforms, *Pseudomonas* spp. and *Candida* spp.) |
| Vandini et al. (2014) [71] | Three independent hospitals—severe brain-damaged and rehabilitation ward, in-patient general medicine ward and geriatric unit (samples collected from corridor floor, room floor, toilet and rehabilitation gymnasiums) | Italy & Belgium | 24 weeks | 50–89% reduction in HAI-related pathogens (*S. aureus*, coliforms, *C. difficile* and *C. albicans*) |
| La Fauci et al. (2015) [70] | University hospital—thoracic and vascular surgical ward (samples collected from corridor floor, inpatient room and dispensary washbasin) | Messina, Italy | 3 months | 92.2–99.9% reduction in pathogen burden (*E. faecalis*, *Pseudomonas* spp., *Acinetobacter* spp., *K. pneumoniae* and *C. albicans*) |
| Caselli et al. (2016) [72] | Private hospital—four randomized rooms located on two floors (samples collected from the floor, bed footboard and bathroom sink) | Ferrara, Italy | 6 months | 98% reduction in bacterial and fungal pathogen burden (*Staphylococcus* spp., *S. aureus*, *Enterobacter* spp., *Pseudomonas* spp., *Clostridium difficile*; *Candida* spp. and *Aspergillus* spp.) Decrease in AMR genes (84 antibiotic resistance genes analyzed) |
| Afinogenova et al. (2017) [78] | Pasteur Institute Medical Centre (samples collected from treatment rooms in the gynecologist office) | Saint Petersburg, Russia | 30 days | Reduction in *Staphylococcus* spp. and Enterobacteriaceae |
| Caselli et al. (2018) [73] | Six public hospitals—general medicine wards (samples collected from the floor, bed footboard and sink) | Italy | 18 months | 83% reduction in pathogen burden (*Staphylococcus* spp., *Enterobacter* spp., *Acinetobacter* spp., *Pseudomonas* spp., *Clostridium difficile*) 59% reduction in HAI-related pathogens 52% decrease in the incidence of HAIs 2 Log decrease in AMR genes (84 antibiotic resistance genes analyzed) |
| Al-Marzooq et al. (2018) [79] | University dental clinic (samples collected from floor, keyboards, spittoon, patient headrest, patient chair, dentist chair, drain, wires of handpieces and sink) | United Arab Emirates | 3 weeks | Reduction in *Staphylococcus* spp. and *Streptococcus* spp. |
| Caselli et al. (2019) [74] | Five public hospitals (samples collected from the floor, bed footboard and bathroom sink) | Ferrara, Italy | 6 months | 99% decrease in AMR genes (84 antibiotic resistance genes analyzed) Decrease in *S. aureus*-resistant strains 60.3% decrease in antimicrobial drug consumption associated with HAIs 75.4% decrease in patientcare associated costs |

**Table 1.** *Cont.*

| Reference | Built Environment | Location | Duration | Outcome (PBS vs. Chemical Disinfectants) |
|---|---|---|---|---|
| D'Accolti et al. (2019) [80] | Private hospital internal medicine ward—four patient rooms (samples taken from bathroom floor, sink, shower plate, room floor and bed footboard) | Ferrara, Italy | 23 days | Reduction in *Staphylococcus* spp. |
| Kleintjes et al. (2020) [81] | Western Cape Provincial—tertiary adult burn unit | Cape Town, South Africa | 3 months | 56% reduction in the incidence of HAIs |
| Soffritti et al. (2022) [77] | Maternal and child health institute—emergency rooms (samples collected from floor, bed footboard and sink) | Trieste, Italy | 9 weeks | 80% reduction in pathogen burden (*Staphylococcus* spp., *Enterobacter* spp., *Acinetobacter* spp., *Pseudomonas* spp., *Clostridium difficile*, *Enterococcus* spp.) 2 Log decrease in AMR genes (84 antibiotic resistance genes analyzed) No detection of SARS-CoV-2 |
| Klassert et al. (2022) [82] | Neurological ward—nine independent patient rooms (samples collected from the floor, door handle and sink) | Berlin, Germany | 3 months | Overall reduction in bioburden (*Staphylococcus* spp., *Streptococcus* spp., *Moraxella* spp. *Enterobacter* spp. and *Veillonella* spp.) Increase in microbial diversity Decrease in AMR genes (12 antibiotic genes analyzed) |
| D'Accolti et al. (2023) [76] | Two Italian hospitals—general medicine wards (samples collected from bathroom floor, sink and shower, room floor and bed footboard) | Rome & Ferrara, Italy | 14 weeks | Reduction in *Staphylococcus* spp. Decrease in *Staphylococcus*-resistant strains |
| Leistner et al. (2023) [83] | University hospital—18 non-ICU wards | Berlin, Germany | 4 months | No change in the incidence of HAIs |
| D'Accolti et al. (2023) [75] | Subway system—two underground driverless trains (samples collected from train floors, seats, handrails, doors and air filters) | Milan, Italy | 12 weeks | Reduction in bacterial and fungal pathogen burden (*Staphylococcus* spp., *Enterobacter* spp., *Pseudomonas* spp., *Clostridium difficile*; *Candida* spp. and *Aspergillus* spp.) Decreased detection of SARS-CoV-2 |

Reference includes the name of the first author, year of publication and reference number; Built Environment identifies the types of environments treated with PBS and the types of environmental surfaces where microbial samples were collected; Location identifies the geographical city and country where study was conducted; Duration identifies length of time of PBS treatment; Outcomes include in situ results observed with PBS treatment, as compared to treatment with chemical disinfectants.

Remarkably, the majority of in situ hospital studies identified in Table 1 have demonstrated that PBS has led to a stable reduction in the pathogen burden on high-touch hospital surfaces as well as a significant reduction in hospital acquired infections (HAIs) [69,73,78,79]. These studies consistently observed significant reductions in total pathogen numbers by the first sampling time point (between 1 and 4 weeks) with PBS treatment, and this reduction was maintained throughout the duration of the experiments [69,72,73,77–79,84]. Alternatively, no significant variations in pathogen numbers were observed at these same time points with chemical disinfectant treatment [69,72,73,77–79,84]. The largest of these studies, an 18-month study conducted across six Italian hospitals in 2016–2017, reported an 83% decrease in the surface pathogen burden and more than a 50% decrease in the number of HAI-associated microorganisms [73]. Similarly, the incidence of HAIs significantly decreased from 4.8 to 2.3%, with nearly 12,000 patients surveyed [73]. This substantial reduction in HAIs was also associated with a 60% decrease in hospital-prescribed antibiotics [74]. Similar results were reported from a geriatric unit, which included elderly patients being treated for infections and HAIs, as well as a tertiary adult burn unit [81].

In addition to the stable reduction in the pathogen burden and associated HAIs, switching from chemical disinfectants to PBS eliminated 50–99.9% of the previously identified resistant bacterial strains on hospital surfaces [74,76]. Abatement of resistant staphylococci strains and other multi-drug resistance bacteria was even more successful when lytic bacteriophages was added to the *Bacillus*-based probiotic cleaning solution [76,80].

PBS treatment was also associated with a significant decrease in the prevalence of AMR genes detected from surface samples across all hospitals [72,73,77,82]. Notably, these declines in the abundance of AMR genes and resistant bacterial strains were not observed with chemical disinfectant use [8]. To the contrary, no statistical differences in HAIs or antimicrobial-resistant bacteria were found across treatment groups when non-ICU wards were treated with PBS, chemical disinfectants or plain soap [83].

To investigate the short-term effects of PBS treatment on surface microbiome composition, Klassert et al. (2022) analyzed the microbial composition of high-touch surfaces in nine patient rooms for three months. Interestingly, the microbial community structures were relatively stable throughout study and the most abundant taxa (non-pathogenic) remained unchanged, regardless of the sanitation method [82]. However, in comparison to chemical disinfectant use, PBS use led to a significant increase in microbial diversity on the surfaces of the hospital sinks and floors [82]. Likewise, the percentage of *Bacillus* detected on high-touch hospital surfaces increased from a median value of 0% to nearly 70% of the total surface microbiota [73]. Destabilization of biofilms on hospital hard surfaces, followed by the replacement with *Bacillus* cells have also been observed with PBS application [71]. Considering that PBS relies on *Bacillus* spp. dominating the surface microbiome composition, human exposure to *Bacillus* spp. will undoubtably increase with PBS use. No AMR genes have been detected in any Bacilli isolates post-PBS application [72,73,77,82], and no PBS-derived *Bacillus* infections have been detected in any of the hospitalized patients [61,63,64,67,68]. Taken together, these results support the idea that PBS operates by a competitive exclusion mechanism that allows for the displacement of resident bacteria and pathogens, and the concomitant replacement with non-pathogenic, non-resistant *Bacillus* spp.

Given the present global interest in cleaning and disinfection to combat the spread of the SARS-CoV-2 virus, Soffritti et al. (2022) replaced conventional chemical disinfectants with PBS in the emergency rooms of a Maternal and Child Health Institute for two months during the COVID-19 pandemic. Similar to previously reported results, PBS was associated with an 80% reduction in HAI-associated pathogens and a significant decrease in the detection of total AMR genes [77]. Notably, while COVID-positive adult and pediatric patients were admitted throughout the study, no positive SARS-CoV-2 surface samples were detected with chemical disinfectant use or PBS use [77]. Despite these overwhelmingly positive results in the hospital environment, the first, and to our knowledge, the only study to-date assessing the impact of *Bacillus*-based PBS in non-healthcare public BEs was recently conducted in an Italian subway system [75]. PBS application led to a nearly 60% reduction in the surface and air pathogen burden within 2 weeks, and to a complete abatement of pathogens by the end of the 12-week study. In contrast, the application of chemical disinfectants was not associated with a reduction in pathogen numbers at any time point during the study [75]. Similarly to PBS use in hospital environments, the reduction in the pathogen burden was associated with a concomitant decrease in the detection of AMR genes. Importantly, a significant decrease in SARS-CoV-2 positive samples were detected in the PBS-treated train, as compared to the train treated with chemical disinfectants [75]. While the size and scope of these studies are limited, taken together the data suggests that PBS is effective in limiting the spread of SARS-CoV-2, and recognizes PBS as a promising, safe, highly effective and eco-sustainable alternative to chemical disinfectants.

## 4. Discussion

Although the number of studies assessing PBS in practice are limited, the results described above and referenced in Table 1 demonstrate the ability of PBS to successfully modulate and stably rebalance the hospital microbiome, reduce pathogen burden and AMR genes and decrease HAIs. Even the least striking results indicate that PBS is just as effective as chemical disinfectants and significantly more health- and environmentally friendly [83]. Accordingly, the potential of also employing PBS in our homes, schools, workplaces and other built environments is compelling and should be investigated. As discussed, cur-

rent evidence demonstrates the effectiveness of an established PBS system in reducing the pathogen burden on high-touch surfaces, including the elimination of antimicrobial-resistant bacteria. However, the limitations and potential unknowns associated with short- and long-term PBS use must also be considered. Most notably, implementation of PBS is accompanied by a lag time that is required to obtain a stable *Bacillus*-dominant surface microbiome. The reports by Caselli et al. indicate that a minimum of two weeks is needed to achieve a surface microbiome that is dominated by *Bacillus* spp., and therefore instantaneous decontamination is not a viable application for PBS [6]. In situations where immediate disinfection is warranted, a chemical disinfectant or alternate disinfection method, such as UV light, steam or hydrogen peroxide products etc. (reviewed by Boyce [85]) would be required. Likewise, once a PBS system is adopted, consistency in application is required to maintain the prophylactic sanitation benefits. Chemical disinfectants can kill *Bacillus* vegetative cells and significantly reduce *Bacillus* spore numbers [75,77,83,86]. Vandini et al. (2014) observed that once PBS protocols were switched back to conventional chemical disinfectant protocols, pathogens numbers increased to pre-PBS levels [71]. The type of target surface material has also been shown to affect the effectiveness of PBS. PBS was less effective on linoleum flooring than on porcelain sinks, and continuous application was required to maintain a significant reduction in the bioburden [70]. Therefore, while the need for regular application is not unique to PBS and is arguably even more necessary for conventional chemical disinfectants to limit recontamination, the initial lag time and inability to freely switch between methods may be a drawback for practical widespread PBS adoption.

The majority of results used to support PBS focus on the reduction in the bacterial pathogen burden, along with a reduction in certain fungal pathogens, on hospital surfaces and the associated decreases in HAIs. Accordingly, an essential question that arises when assessing PBS's usefulness is its effectiveness in eliminating viral threats. Recent in vitro results demonstrate that PBS is able to effectively inactivate all tested enveloped viruses within 1–2 h of contact time, including hCoV-229E and SARS-CoV-2 coronaviruses, human herpesvirus type 1, human and animal influenza viruses and vaccinia virus [87]. Notably, the antiviral action of PBS was just as effective as the control chemical disinfectants and persisted significantly longer—24 h, compared to only 2 h with chemical disinfectants [87]. While these results are encouraging, viral infectivity is vast—evaluating the effectiveness of PBS on the more resistant non-enveloped viruses is necessary to have the desired confidence to employ PBS in high-risk settings.

While PBS is promising, the use of PBS in built environments is in its infancy and several unknowns still remain. All published in situ evidence supports the claim that non-pathogenic *Bacillus* spp. are safe for humans in a variety of applications, including water treatment, agriculture, livestock, food preparation and gut homeostasis [88–93]. In vitro assessment of airborne *Bacillus* levels in indoor environments that were treated with *Bacillus*-based carpet cleaners estimated minimal risk associated with inhalation of *Bacillus* spores and cells [94]. Consistent with these results, all of the studies discussed here support the claim that these probiotic bacteria pose no health risks to even the most vulnerable individuals [71,72,77]. However, several recent studies have identified transferable AMR genes in commercial animal and human probiotics, including *Bacillus* spp. [66,95,96]. Probiotic *Bacillus* spp. have also been observed to produce biogenic amines and harmful enterotoxins [97]. Widespread use of PBS in built environments would substantially increase human exposure to *Bacillus* spores and vegetative cells and would warrant long-term studies to evaluate the potential ecological impact of PBS use on the human microbiota and the effects on human health.

Targeting pathogens while maintaining commensal microbial diversity is the ultimate goal of bidirectional hygiene, and recent evidence indicates PBS is much more effective at achieving this goal than conventional chemical-based methods. Interestingly, Stone et al. (2020) demonstrated that plain soap was more effective than both chemical disinfectants and PBS in maintaining microbial biodiversity on surfaces [47]. Moreover, in vitro

data have shown that over time, *Bacillus* spp. can compete with each other, as well as common environmental bacterial species, which may hinder their ability to persist on applied surfaces [98–100]. Accordingly, while *Bacillus* spp. are the most widely used and studied probiotics in microbial-based cleaning products, expanding the microbial diversity within these products has been recommended [91]. Early in vitro studies assessing the effectiveness of *Lactobacillus* spp., particularly *L. lactis* and *L. acidophilus,* have demonstrated that these species can reduce the adhesion of pathogenic bacteria to laboratory hard surfaces [101–103]. Likewise, the application of *Streptococcus thermophilus* and *Streptococcus mitis* have been shown to significantly reduce the in vitro colonization of *S. aureus* and *C. albicans* on rubber, silicone and glass surfaces [101,104]. Thus, investing in the development and evaluation of diverse PBS consortiums and formulations, such as those containing bacteriophages [76,80,91], lactic acid bacteria and photosynthetic microbes [105], could prove worthwhile, as long as quality control and assurance is upheld. An analysis of 14 commercial probiotic cleaners identified the presence of several opportunistic bacterial pathogens in the product preparations [106]. Stringent testing must also be conducted throughout the manufacturing process to ensure accurate probiotic counts, as large variations in bacterial counts have been observed among commercial probiotic cleaners—of which the lowest concentrations identified would likely render the product ineffective at directed use concentrations [106].

Although there are numerous microbial-based cleaners on the market, there is currently no mandatory regulatory oversight to govern probiotic microorganisms in cleaning products [107]. In the United States, the probiotic species used in these commercial cleaners are considered food grade with a GRAS label ("generally recognized as safe"), therefore there are no additional regulations required to evaluate efficacy and safety [45,107]. Likewise in Europe, only legislation on the occupational safety of biological agents applies to microbial-based cleaning products [107]. Consequently, recent studies have indicated that toxicological risk assessments, hygienic practices and quality control significantly varies among product manufacturers [66,91,107]. Voluntary ecolabelling has become increasingly popular [108], yet certification is limited to verifying human safety, product efficacy and environmental preferability; thus, precise identification and concentration of the microbial consortium is not necessary for the certification. The taxonomic genera tend to be the only information provided on product labels, as manufacturers consider the precise identity and composition as proprietary and confidential [109,110].

Lastly, to appeal to the general population and decision makers, PBS will need to be economically attractive to be utilized on a larger scale. A six-month Italian study found that PBS use was associated with a 60% decrease in antimicrobial drug consumption and a 75% decrease in associated HAI-related costs [64]. Tarricone et al. (2020) performed a five-year budget impact analysis comparing the estimated cost/savings of PBS versus conventional chemical cleaning in Italian hospital departments [111]. The analysis found that the increased utilization of PBS, rather than chemical disinfectants, would prevent over 30 thousand HAIs and 8500 antibiotic-resistant infections and save an estimated 14 million euros [111]. Given the limited number of studies that have investigated PBS in practice, as well as the limited locations and environments represented, similar budget analyses will be vital if, or when, PBS is implemented in additional countries, settings and on a larger scale.

## 5. Conclusions

Taken together, the evidence discussed here supports reducing our reliance on, and exposure to, chemical disinfectants and encourages a shift towards implementing more holistic and sustainable sanitation approaches that eliminate pathogens yet maintain a beneficial microbiome in our built environments. The potential direct and indirect benefits of widespread PBS use on short- and long-term individual and public health are tremendous, which only stresses the need to tackle the remaining unknowns and challenges. Mandatory regulatory standards and consistent ecolabelling practices that provide assurance and guarantee product safety and efficacy are essential. Public health awareness campaigns

from authorized sources that focus on bidirectional hygiene and hygiene education are recommended to increase public understanding of the need for and benefits of this paradigm shift. Last, but certainly not least, continued scientific investigations into PBS in practice, in diverse built environments and with extended study durations, are needed to evaluate widespread PBS use and the impact on the health of our built environments.

**Author Contributions:** Conceptualization, A.L.F.; methodology, A.L.F.; investigation, A.M.R. and A.L.F.; resources, A.L.F.; data curation, A.L.F.; writing—original draft preparation, A.L.F.; writing—review and editing, A.M.R. and A.L.F.; visualization, A.L.F.; supervision, A.L.F.; project administration, A.L.F. All authors have read and agreed to the published version of the manuscript.

**Funding:** This research received no external funding.

**Data Availability Statement:** No new data were created or analyzed in this study. Data sharing is not applicable to this article.

**Acknowledgments:** Thank you to Donna Hamilton, Kelly Varga and Corron Sanders at the University of North Texas at Dallas for their insightful discussions.

**Conflicts of Interest:** The authors declare no conflict of interest.

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
