# Peer review of "Probiotic-Based Sanitation in the Built Environment—An Alternative to Chemical Disinfectants"

_2673-8007, doi:10.3390/applmicrobiol3020038_

Round 1

Author Response

Thank you for your feedback. Please see the attachment.

Reviewer 2 Report

Dear Authors,

The article addresses an interesting topic from a scientific point of view, an effective method experienced through in situ studies, to prevent health care associate infections and even reduce their incidence. The article is well written and structured and requires only minor changes.

 Suggestions:

Introduction.

Line 96 – “BEs” – please, explain the abbreviation, since is the first-time when you use this term.

It would be useful to introduce an original scheme to explain the PBS hypothesis.

 Materials and methods

Add some selection criteria for the 15 articles considered as primary research articles

Author Response

(The authors gave the same response as above.)

Reviewer 3 Report

General comments

In the introduction, it would be good for the authors to also indicate specific chemical disinfectants and the resistance reported to be associated with them. This should include the antibiotic cross-reactivity mentioned.

Regarding the other disinfectants, it would be good to also mention the nonbiological disadvantages (e.g. corrosion) of these, like chlorine for example. 

While it is clear that the authors have mentioned that some of these disinfectants only last a few hours or days, it would be good to give examples that give specific durations. Please include the percentage effectiveness of these chemical disinfectants.

It may also be beneficial for the authors to point out the effectiveness in the action of probiotics. Although the Table gives certain percentages, it is not clear how long it took to achieve these percentage reductions.

Finally, several studies have reported a growing concern for antibiotic resistance genes and mobile genetic elements in probiotic bacteria. It would therefore be helpful if the authors could provide any literature on the possibility of such happening with probiotic disinfectants

Specific comments

Lines 47-48: ....abundance of antimicrobial resistance genes and antibiotic-resistant organisms

English language is fine. Minor edits are required

Author Response

(The authors gave the same response as above.)

Round 2

Reviewer 3 Report

Than you for addressing the comments and improving the manuscript.

English language is fine